# DTFusion: Infrared and Visible Image Fusion Based on Dense Residual PConv-ConvNeXt and Texture-Contrast Compensation

**DOI:** 10.3390/s24010203

**Published:** 2023-12-29

**Authors:** Xinzhi Zhou, Min He, Dongming Zhou, Feifei Xu, Seunggil Jeon

**Affiliations:** 1School of Information, Yunnan University, Kunming 650504, China; xzzhou163@163.com (X.Z.); zhoudm@ynu.edu.cn (D.Z.); xufeifei@stu.ynu.edu.cn (F.X.); 2Samsung Electronics Co., Ltd., 129 Samseong-ro, Yeongtong-gu, Suwon-si 16677, Republic of Korea; simon.sgjeon@gmail.com

**Keywords:** attention mechanism, deep learning, gradient residuals, image fusion, infrared and visible, larger receptive fields

## Abstract

Infrared and visible image fusion aims to produce an informative fused image for the same scene by integrating the complementary information from two source images. Most deep-learning-based fusion networks utilize small kernel-size convolution to extract features from a local receptive field or design unlearnable fusion strategies to fuse features, which limits the feature representation capabilities and fusion performance of the network. Therefore, a novel end-to-end infrared and visible image fusion framework called DTFusion is proposed to address these problems. A residual PConv-ConvNeXt module (RPCM) and dense connections are introduced into the encoder network to efficiently extract features with larger receptive fields. In addition, a texture-contrast compensation module (TCCM) with gradient residuals and an attention mechanism is designed to compensate for the texture details and contrast of features. The fused features are reconstructed through four convolutional layers to generate a fused image with rich scene information. Experiments on public datasets show that DTFusion outperforms other state-of-the-art fusion methods in both subjective vision and objective metrics.

## 1. Introduction

Infrared and visible images for the same scene contain different information due to different imaging mechanisms of the sensors. In general, visible images have higher resolution and richer texture details. However, visible image sensors are susceptible to light intensity and weather factors, in which it is hard to distinguish prominent scene targets. Infrared sensors can overcome the impacts of dark environments and weather changes to capture the target thermal radiation information, but infrared images have poor contrast and blurred background textures. Thus, infrared and visible image fusion attempts to integrate the prominent target from an infrared image with the texture information from a visible image to produce a fused image that contains the complementary information from source images. Currently, infrared and visible image fusion techniques [1] are widely used in many fields, such as image enhancement [2], object segmentation [3], target detection [4], and fusion tracking [5].

In recent years, scholars have proposed numerous fusion methods, which can be classified into traditional methods and deep-learning-based methods. Among the traditional fusion methods, typical methods include multiscale transform [6,7], sparse representation [8,9], saliency fusion [10,11], subspace analysis [12,13] and other hybrid methods [14,15]. However, traditional fusion methods tend to ignore the differences of image features and extract feature information indiscriminately, which results in the loss of details and characteristic information of the source images. In addition, the feature extraction mechanisms and fusion strategies are increasingly complex and time-consuming.

The powerful learning ability of deep learning has compensated for the limitations of traditional methods to some degree, such as convolutional neural network (CNN)–based methods [16,17,18], autoencoder (AE)-based methods [19,20,21], generative adversarial network (GAN)–based methods [22,23], and Transformer-based methods [24,25]. CNN-based methods generally implement feature extraction, fusion, and image reconstruction by carefully designing the loss function to train the network. AE-based methods can train autoencoders on large natural datasets for feature extraction and reconstruction, but most of the fusion strategies are handcrafted and cannot achieve end-to-end learning. GAN-based methods mainly establish an adversarial mechanism between generators and discriminators to enforce the generation of the desired fused images. Although it avoids manually designing fusion strategies, a large amount of detail information is lost during the adversarial training process. In addition, these methods extract feature information within a limited receptive field by stacking small kernel-size convolutions, which inevitably ignores some global information. Recently, Transformer has been employed in infrared and visible image fusion tasks due to its global feature extraction capabilities. Transformer-based methods have demonstrated significant fusion performance. However, some methods simply stack Transformer blocks, while ignoring the importance of local feature information, or fail to account for differences between different modal images, which results in information redundancy.

To overcome the above challenges, a novel end-to-end network without a separate manually designed fusion layer is proposed to fuse infrared and visible images, which is called DTFusion. Partial convolution (PConv) [26] is introduced to enhance the ability of extracting spatial features of the ConvNeXt block [27] so that the constructed PConv-ConvNeXt block (PCB) with large kernel-size convolution can extract features from a larger reception field efficiently. Then, dense connections are applied between PConv-ConvNeXt modules (RPCMs) to reuse intermediate features. Furthermore, a texture-contrast compensation module (TCCM) with gradient residuals and an attention mechanism is used to compensate for texture details and contrast of features, so that fused images are generated by integrating these features. The main contributions are as follows:In order to extract features from larger receptive fields more efficiently, PConv is introduced into the ConvNeXt block, and we construct an RPCM with large-kernel-size convolution. Then, dense connections are applied between RPCMs to reuse intermediate features.To integrate more complementary features to the fusion result, TCCM with gradient residual connections and an attention mechanism is designed to compensate for texture details and contrast of features.Extensive experiments on different test datasets are implemented, and the results show the excellent fusion performance of the suggested method on subjective and objective metrics compared with other state-of-the-art deep-learning-based methods.

The rest of the paper is organized as follows: Section 2 focuses on the development of image fusion methods and background information on the ConvNeXt block. Section 3 describes the principles of DTFusion. Ablation studies and comparative experiments are presented in Section 4. Finally, Section 5 concludes with relevant conclusions.

## 2. Related Work

In this section, we introduce the development of deep-learning-based methods and describe the details of the ConvNeXt block.

### 2.1. Image Fusion Method Based on Deep Learning

In recent years, a variety of deep-learning-based fusion algorithms have emerged in the field of image fusion. Li and Wu [19] proposed an encoder–decoder fusion network of DenseFuse by employing convolutional layers and dense blocks into the encoder network to replace the traditional CNN representation model, which uses addition and $L_{1}$-norm as fusion strategies to fuse the infrared and visible images adaptively. However, merely using dense blocks would result in a loss of detail during the convolution process. To extract more comprehensive features, Li et al. [20,28] further designed NestFuse and RFN-Nest. The former extracted multiscale features from the source image by using nest connection in the encoder network. Based on the former, the latter presented a learnable residual fusion network that achieves end-to-end fusion. Wang et al. [29] proposed a multiscale densely connected fusion network called UNFusion, which develops an attentional fusion strategy based on $L_{1}$, $L_{2}$, and $L_{\infty }$ norms to combine multiscale features. Considering the information redundancy brought by multiscale features, Jian et al. [30] suggested a symmetric encoder–decoder framework with residual block fusion called STDFusionNet, which adopts intermediate feature compensation and attentional fusion to preserve the salient target and texture details from a source image. However, these methods either train autoencoders on large natural datasets (MS-COCO) and use undifferentiated convolutional kernel parameters for feature extraction from infrared and visible images, or manually design fusion strategies additionally that cannot be learned by the network, which inevitably lowers the performance of network fusion.

Compared with the above methods, Ma et al. [22] proposed a GAN-based fusion model called FusionGAN, which establishes an adversarial mechanism between the fused image and the source image to preserve the rich texture information. However, a single discriminator tends to cause fusion imbalance and fails to make the fused image photorealistic. To solve this problem, they further designed a GAN with multiple classification constraints (GANMcC) [23]. However, GAN is difficult to train stably, which is unable to preserve the rich detail information into the fused image. Xu et al. [18] developed U2Fusion, which is a unified fusion framework that fuses different types of images by measuring information preservation and training a model with elastic weights. Since U2Fusion is not specifically designed for infrared and visible image fusion tasks, it did not achieve excellent performance.

### 2.2. Image Fusion Method Based on Transformer

Unlike CNNs, Transformer is capable of modeling long-range dependencies between image features. Consequently, Transformer-based fusion models have received widespread attention. Wang et al. [24] proposed a full-attention feature coding network with a pure Swin Transformer to capture a global context; however, it ignores representing the local information. To comprehensively focus on both global and local information, Chen et al. [25] developed THFuse, in which a two-branch CNN module is utilized to extract shallow features, and then a visual Transformer module is introduced to model the global relationships of the features. Tang et al. [31] designed a dual-attention Transformer fusion network to examine the important regions of source images and preserve global complementary information. However, in these methods, infrared and visible images are simply spliced to model global relationships without considering modal differences, which inevitably leads to information redundancy.

Therefore, an end-to-end framework is proposed in this article to alleviate the problems. First, RPCM with large kernel-size convolution is designed to extract features from a larger reception field efficiently. Second, gradient residuals and an attention mechanism are used to compensate for texture details and contrast of features. In addition, the network is trained on a dataset consisting of both infrared and visible images, which allows for more targeted feature extraction.

### 2.3. ConvNeXt Block

Since a Transformer model based on the self-attentive mechanism lacks the inherent inductive biases of CNNs, Liu et al. [27] proposed a pure ConvNet model named ConvNeXt. This model maintains the simplicity and efficiency of the standard ConvNet and can compete with Transformers in computer vision tasks. The structure of the ConvNeXt block is shown in Figure 1.

The first layer of the ConvNeXt block is a 7 × 7 depthwise convolution (DWConv) layer, which performs a separate convolution operation for each channel of the feature map, similar to the weighted sum operation in self-attention. The DWConv layer is followed by a layer normalization (LN) [32], which operates on a single sample at a time. Next, the channel dimension of the feature map is expanded by a factor of four through a 1 × 1 convolution layer and activated with the Gaussian error linear unit (GELU) [33] function. Then, the channel dimension of the feature map is reduced by a factor of four through a 1 × 1 convolution layer. Such a design increases the network width, which is more effective in extracting feature information. Finally, the output from the ConvNeXt block is obtained by adding the input and output through a skip connection.

In our work, by introducing PConv into the ConvNeXt block, a PCB is constructed to design for the proposed RPCM that servers as the encoder network, which consists of a convolutional layer and residual connections of three PCBs. PConv can effectively extract spatial features and further compensate for the potential accuracy drop caused by DWConv.

## 3. Our Method

The overall processes of DTFusion is shown in Figure 2, which uses an encoder–decoder architecture. The encoder contains two separate feature extraction channels for infrared and visible images, each consisting of a convolutional layer, two RPCMs, and a TCCM. The convolutional layer with a kernel size of 3 × 3 is used to extract shallow features. RPCM efficiently extracts deep features with larger receptive fields, and dense connections are applied between RPCMs to reuse intermediate features. Then, TCCM is used to compensate for texture details and contrast of features. The decoder consists of four concatenated 3 × 3 convolutional layers that are used to integrate complementary information and generate the fused image.

### 3.1. Residual PConv-ConvNeXt Module

The structure of our designed RPCM is depicted in Figure 3, which consists of a convolution layer and residual connections of three PCBs. Each PCB is constructed by integrating a PConv into the first layer of the ConvNeXt block. Specifically, one quarter of the channels for the input features are subjected to a 7 × 7 regular convolution operation, while the remaining channels are subjected to a 7 × 7 DWConv operation. Since DWConv reduces the accuracy, a pointwise convolution is usually used after DWConv to compensate for the drop of accuracy by increasing the number of channels. Therefore, the accuracy of the spatial features that are extracted by the ConvNeXt block can be further compensated by PConv.

First, a convolutional layer is used to change the number of channels for the input feature *F* and the output feature $F_{0}$. Then, the output $F_{RPCM}$ of RPCM is obtained from $F_{0}$ through the residual connection of three PCBs. They are expressed as follows: (1)$$F_{0}=Conv{}^{3\times 3}\left(F\right)$$
(2)$$F_{RPCM}=F_{0}+{PCB}_{m}\left(F_{0}\right)$$
where $Conv{}^{3\times 3}$ represents a 3 × 3 convolution and $PCB_{m}$ denotes *m* PCBs; *m* is set 3 in this paper.

### 3.2. Texture-Contrast Compensation Module

As shown in Figure 4, TCCM consists of two gradient residuals with SimAM [34] attention and a channel and spatial attention module (CSAM). The Sobel and Laplace operators are used to compensate for the fine-grained representation of features. CSAM assigns higher weights to features with higher contrast.

First of all, for the input original feature map *F*, the output $F_{TCCM}$ of the texture-contrast compensation is given by
(3)$$F_{TCCM}=Concat\left(CSAM\left(F\right)\;⊕\;S_{AM}\left(\nabla^{2}\left(F\right)\right),S_{AM}\left(\nabla \left(F\right)\right)\right)$$
where Concat(·) represents the concatenation operation on the channel dimension, CSAM(·) indicates channel and spatial attention operations, ⊕ denotes element summation, $S_{AM}\left(\cdot \right)$ stands for SimAM attention operations, ∇ and $\nabla^{2}$ denote the Sobel and Laplace operators, respectively. The Sobel operator is used to calculate the gradient of the image intensity at each pixel, which can preserve the strong texture features. The Laplace operator is employed to detect edges and fine details of the image, which contributes to further extracting the weak texture features. The common fine texture is enhanced by adding output features from CSAM and weak texture features via an element-wise addition, while stronger texture details are preserved by splicing the channels with strong texture features. Finally, a more complete representation for texture features can be obtained by combining the two gradient residuals. In addition, the SimAM attention module can assign higher weights to salient texture features.

Next, in CSAM, the input feature map *F* is subjected to global average pooling and maximum pooling in the spatial dimension. Then, the pooling results $F_{Avg}^{C}$ and $F_{Max}^{C}$ are fed into the multilayer perceptron (MLP) for learning, and the outputs of the MLP are summed and activated by a sigmoid function to generate a channel attention weight map $M_{C}\in R^{C\times 1\times 1}$. Finally, the channel attention feature map $F_{C}$ is obtained by multiplying the input *F* with the weight map $M_{C}$. This process can be represented as follows: (4)$$M_{C}\left(F\right)=\sigma \left(MLP\left(F_{Avg}^{C}\right)+MLP\left(F_{Max}^{C}\right)\right)$$
(5)$$F_{C}=M_{C}\left(F\right)⊗F$$
where σ represents the sigmoid function, MLP(·) indicates multilayer perceptron operation, and ⊗ denotes element-wise multiplication. Similarly, the input feature map $F_{C}$ is subjected to global average pooling and maximum pooling along the channel axis. Then, the pooling results $F_{Avg}^{S}$ and $F_{Max}^{S}$ are concatenated by channel dimension for the convolution operation to generate the spatial attention weight map $M_{S}\in R^{1\times H\times W}$. The final attention feature map $F_{O}$ is produced by multiplying the input feature map $F_{C}$ with the weight map $M_{S}$. The specific formulation is as follows: (6)$$M_{S}\left(F\right)=\sigma \left(Conv^{7\times 7}\left(Concat\left(F_{Avg}^{S},F_{Max}^{S}\right)\right)\right)$$
(7)$$F_{O}=M_{S}\left(F_{C}\right)⊗F_{C}$$
where $Conv{}^{7\times 7}$ represents a 7 × 7 convolution.

### 3.3. Loss Function

By using a loss function to optimize the fusion network, more visible texture details can be preserved in the fused image, and the infrared thermal information can be highlighted. Therefore, we apply structural similarity (SSIM) [35] loss, intensity loss, and texture loss to jointly constrain the network.

#### 3.3.1. SSIM Loss

The SSIM metric measures the similarity of two images in terms of brightness, contrast, and structure. We employ the SSIM loss function to force the fused image to retain more structural information from the two source images. The SSIM loss function is defined as
(8)$$L_{ssim}=w_{1}\cdot \left(1-ssim\left(I_{f},{I_{i}}_{r}\right)\right)+w_{2}\cdot \left(1-ssim\left(I_{f},I_{vis}\right)\right)$$
where ssim(·) represents the SSIM operation, and $I_{ir}$, $I_{vis}$, and $I_{f}$ denote the two input source images and the output fused image, respectively. In general, visible images have sharper texture detail than infrared images in daytime scenes. Conversely, infrared images provide more prominent targets and richer texture than visible images in nighttime scenes. To handle complex scenes and to better balance the fused information, structural information from both source images is equally important. Therefore, we set the parameter scale $w_{1}$ = $w_{2}$ = 0.5.

#### 3.3.2. Intensity Loss

The fused image should maintain the optimal brightness of the highlighted target. For this purpose, we utilize an intensity loss function to integrate the pixel intensity information from the source images: (9)$$L_{int}=\frac{1}{HW}{∥I_{f}-\max\left(I_{ir},I_{vis}\right)∥}_{1}$$
where *H* represents the height of the image, *W* represents the width of the image, max(·) refers to the maximum selection by element, and ${∥\cdot ∥}_{1}$ denotes the $L_{1}$-norm.

#### 3.3.3. Texture Loss

The fused image is expected to characterize the global intensity information and to contain more texture information simultaneously. Hence, the texture loss is used to integrate more texture details of the source image into the fused image: (10)$$L_{text}=\frac{1}{HW}{∥\left|\nabla I_{f}\right|-\max\left(\left|\nabla I_{ir}\right|,\left|\nabla I_{vis}\right|\right)∥}_{1}$$
where $\left|\cdot \right|$ represents the absolute operation.

Finally, the total loss function is defined: (11)$$L_{total}=\lambda_{1}L_{ssim}+\lambda_{2}L_{int}+\lambda_{3}L_{text}$$
where $\lambda_{1}$, $\lambda_{2}$, and $\lambda_{3}$ are the hyperparameters to balance the loss function.

### 3.4. Fusion Evaluation

The performance of the fusion network is often evaluated by subjective and objective methods. The subjective method mainly relies on human eye observation to capture the perceptual quality of the fused image so as to make an evaluation based on the visual effect comparison between the source image and the fused image.

However, it is one-sided to evaluate network performance only by subjective vision. Therefore, objective metrics are introduced to obtain a comprehensive evaluation. We selected nine objective metrics to evaluate the performance, and these metrics are entropy (EN) [36], spatial frequency (SF) [37], average gradient (AG) [38], standard deviation (SD) [39], sum of correlation differences (SCD) [40], visual information fidelity (VIF) [41], gradient-based fusion performance (Q${}_{abf}$) [42], structural similarity index metric (SSIM) [35], and multiscale structural similarity index metric (MS-SSIM) [43]. The higher the value of each above metrics, the better the performance.

## 4. Experiment and Analysis

### 4.1. Experimental Design

#### 4.1.1. Dataset

The MSRS dataset [44,45] contains 1444 pairs of strictly aligned infrared and visible images, of which 1083 training image pairs and 361 test image pairs were used for training and evaluating our fusion task, respectively. Furthermore, 20 image pairs from the TNO [46] dataset and 300 image pairs from the M${}^{3}$FD [47] dataset are selected for ablation analysis and generalization performance verification.

#### 4.1.2. Training Details

The training images are cropped into 64 × 64 patches to expand the dataset. And ultimately, 26,112 pairs of images are used for training. In addition, all images are converted to the grayscale range [0, 1]. The model parameters are updated through the Adam optimizer, the learning rate is initialized to 0.001 and decayed exponentially, and the batch size and epoch are set to 64 and 4, respectively. The hyperparameters of the balanced loss function are set to $\lambda_{1}$ = 10, $\lambda_{2}$ = 10, and $\lambda_{3}$ = 20. All experiments are performed on an Intel i9-10900F 9700 CPU, NVIDIA GeForce RTX 3090 24 GB GPU and 32 GB of RAM. The PyTorch 2.0 platform is used for the proposed program.

#### 4.1.3. Test Details

The testing phase was performed on three datasets—MSRS, TNO, and M${}^{3}$FD. We compare our model with other nine current state-of-the-art deep-learning-based methods, namely, DenseFuse [19], CSF [21], RFN-Nest [28], SDNet [17], GANMcC [23], U2Fusion [18], SMoA [48], SeAFusion [49], UNFusion [29], and DATFuse [31].

### 4.2. Ablation Analysis

Ablation experiments were performed on 20 image pairs selected from the TNO dataset. The contributions, such as loss functions, RPCM, PConv, and TCCM, in improving the network performance were verified through subjective and objective analyses. Additionally, the detailed comparisons can be found in the following figures and tables. Through a large number of experiments with multiple balanced hyperparameters, i.e., $\lambda_{1}$, $\lambda_{2}$, and $\lambda_{3}$, we find that our model achieves the best fusion performance under $\lambda_{1}$ = 10, $\lambda_{2}$ = 10, and $\lambda_{3}$ = 20. However, it is tedious to discuss the process of tuning parameters since there are three hyperparameters in the fusion network. Thus, in this paper, we omit the discussion of how to set hyperparameters in detail.

#### 4.2.1. Ablation Analysis of Loss Function

In our method, structural similarity loss, intensity loss, and texture loss are applied to balance the fused images. To illustrate the necessity of the loss function, we trained the fusion network by eliminating structural similarity loss (Without-ssim), intensity loss (Without-int), and texture loss (Without-text). The subjective comparison results of different loss functions on the two images, *Marne_04* and *Bunker*, are shown in Figure 5, where the prominent targets are marked in a red box and the area in the green box is enlarged to better show the texture details. Although Without-ssim achieves good visual performance, it fails to maintain an optimal structure and intensity information. In contrast, ours produced sharper window contours and cloud edges in *Marne_04* and brighter infrared targets in *Bunker*. Without-int preserves richer texture details, but infrared thermal information is lost. In addition, Without-text lacks visible texture detail. Especially, the bushes in *Bunker* and the window contours in *Marne_04* are blurred.

The results of the objective evaluation metrics are shown in Table 1. The best averages are marked in bold, and the second best averages are underlined. Ours performs best on EN, SD, SCD, SSIM, and MS-SSIM, and achieves suboptimal results on SF, AG, VIF, and Q${}_{abf}$. From the perspective of multimetric evaluation, our results achieved the best performance among the nine objective evaluation metrics.

#### 4.2.2. Ablation Analysis of Network Structure

To verify the effectiveness of the proposed RPCM, PConv, and TCCM structures, we compare with three modified models by replacing RPCM with regular convolution (No-RPCM), eliminating PConv from RPCM (No-PConv), and eliminating TCCM (No-TCCM). The subjective comparison results of different network structure on two images, *Sandpath* and *Heather*, are shown in Figure 6, where the prominent targets are marked in a red box and the area in the green box is enlarged to better show the texture details. No-RPCM preserves the brightness of the infrared target and the visible details. However, when comparing the tree trunk in *Sandpath* and the railing in *Heather*, our DTFusion shows finer texture details and higher contrast. No-PConv method performs close to DTFusion, but ours has higher contrast on the scenes of treetops below the railing in *Heather*. Additionally, although No-TCCM preserves the brightness of the infrared pedestrian target, it lacks visible texture details; e.g., tree trunks and railings are unclear.

Table 2 shows the results of nine objective evaluation metrics. Ours achieves the optimal results on EN, SF, SD, SCD, and MS-SSIM and the suboptimal results on AG. The performance is consistent with the subjective evaluation.

### 4.3. Comparative Experiment

The MSRS dataset contains two typical scenes, daytime and nighttime. Therefore, we selected two daytime and two nighttime scenes for subjective evaluation. In daytime scenes, the fusion result should contain rich visible texture details and infrared salient targets. As shown in Figure 7 and Figure 8, where the prominent targets are marked in a red box and the area in the green box is enlarged to better show the texture details. DenseFuse, RFN-Nest, and SMoA can preserve the visible texture details, but the brightness of these pedestrian targets is lost. However, CSF, SDNet, U2Fusion, and GANMcC preserve the brightness of the infrared pedestrians, but the visible details are unclear or even missing. In particular, GANMcC produces some blurring at the edges of thermal radiation targets, and the overall scene is dark for CSF, SDNet, and U2Fusion. DATFuse preserves the brightness and texture information from visible images, but it fails to capture the brightness and detail from infrared images. Although SeAFusion and UNFusion preserve visible details and brightness information, DTFusion not only has comprehensive scene information, but also preserves richer texture details and higher contrast.

In nighttime scenes, the visible image typically contains limited detail information. However, infrared images contain prominent targets and some background information. As shown in Figure 9 and Figure 10, where the prominent targets are marked in a red box and the area in the green box is enlarged to better show the texture details. DenseFuse, CSF, and RFN-Nest can preserve the visible texture details, but the brightness of these pedestrian targets is lost. SDNet and U2Fusion fail to render road markings and fences hidden in the dark. Although GANMcC retains the highlighted pedestrians, the outlines of the target are blurred. SMoA only achieves limited visual effects. DATFuse obtains overall brightness and rich texture detail, but fails to highlight pedestrian brightness. Except for SeAFusion, UNFusion, and DTFusion, none of these methods can preserve the brightness of pedestrians and the details of road markings and fences simultaneously.

Objective metrics are used to further validate the proposed DTFusion. Table 3 presents the comparison results of nine objective evaluation metrics with nine other methods for 361 image pairs from the MSRS test dataset. DTFusion achieves the optimal averages for SF, AG, SD, SCD, SSIM, and MS-SSIM and the suboptimal averages for EN and Q${}_{abf}$. The best results for SF and AG show the superior ability of the proposed method to preserve detail and sharpness. The maximum SD indicates that the fusion results have high spatial contrast. The optimal SCD shows that the fused image has similar features to the source image. The optimal results for SSIM and MS-SSIM indicate that the proposed method preserves rich structural texture. The suboptimal results for EN and Q${}_{abf}$ imply a relatively excellent ability to fuse visual information and edge information. However, the value of VIF is lower than that of SeAFusion and UNFusion. The possible reason is that the structural and contrast information of the two source images is of equal importance for the fusion result in the SSIM loss function, which may result in a lower fidelity of visual information. A comprehensive evaluation combining several objective metrics showed that DTFusion achieves the best fusion performance, which is consistent with the subjective evaluation results.

### 4.4. Generalization Experiment

We evaluate the generalization of DTFusion on the TNO and M${}^{3}$FD datasets.

#### 4.4.1. Experiments on TNO Dataset

The generalizability of DTFusion was evaluated on 20 image pairs selected from the TNO dataset. The subjective comparison results of different fusion methods on two images, *Kaptain_1654* and *soldier_in_trench_2*, are shown in Figure 11 and Figure 12, where the prominent targets are marked in a red box and the area in the green box is enlarged to better show the texture details. DenseFuse, RFN-Nest, and U2Fusion preserve the visible texture details, but the overall brightness information and the infrared salient targets are lost. Although SDNet and SMoA can preserve the infrared salient features, the visible texture details are missing. GANMcC retains the infrared thermal targets with blurred boundaries and severe loss of texture detail. The ability of CSF to preserve luminance information and UNFusion to preserve texture details is still limited. DATFuse fails to maintain infrared salient luminance and loses some important visible texture details. In contrast, SeAFusion and DTFusion have brighter scenes and texture details. In addition, DTFusion preserves details better than SeAFusion; for example, the contours of the bench under the shack and the branches above the trench are clearer.

The objective comparison results of the different fusion methods on *Kaptain_1654* and *soldier_in_trench_2* are shown in Table 4 and Table 5. For *Kaptain_1654*, DTFusion achieves the optimal metrics for SF, AG, SD, SCD, Q${}_{abf}$, and MS-SSIM and the suboptimal metrics for EN. For *soldier_in_trench_2*, DTFusion obtains the optimal metrics for AG and Q${}_{abf}$ and the suboptimal metrics for SF and VIF. For a more visual analysis, Table 6 shows the objective average results for the 20 image pairs. DTFusion achieves the optimal results for SF, AG, SCD, Q${}_{abf}$, and SSIM and the suboptimal results for EN, VIF, and SSIM. The overall results show that our method retains more detailed structural textures and edge information, higher clarity, and richer information compared with the other nine fusion methods.

#### 4.4.2. Experiments on M${}^{3}$FD Dataset

The generalization capability of DTFusion was further validated on 300 image pairs selected from the M${}^{3}$FD dataset. A representative example of different fusion methods is shown in Figure 13, where the prominent targets are marked in a red box and the area in the green box is enlarged to better show the texture details. DenseFuse, CSF, RFN-Nest, and U2Fusion lose the brightness of pedestrian targets. The contours of the traffic lights are unclear when adopting SMoA and DATFuse. However, SDNet, GANMcC, and UNFusion still have limited ability to retain the structures of the traffic lights and the brightness of the pedestrians. Only SeAFusion and DTFusion obtain excellent visual results, and the structural textures of the traffic lights are more clearly presented in DTFusion. Table 7 shows the objective average results for the 40 image pairs. DTFusion obtains optimal metric for SF, AG, SD, VIF, and Q${}_{abf}$, and suboptimal metric for EN. Overall, the excellent visual effect and higher evaluation metrics on the three different datasets show that DTFusion achieves better fusion performance.

### 4.5. Computational Efficiency Comparison

To evaluate the computational performance of different fusion methods, we compare the average running time of different fusion methods on the three public datasets. All methods were running on a similar machine configured with an Intel i9-10900F 9700 CPU and an NVIDIA GeForce RTX 3090 graphics processor. As shown in Table 8, SDNet and DATFuse perform best. That is because SDNet uses a 1 × 1 convolutional kernel and has a smaller number of feature channels, which allows for high-speed fusion. DATFuse has a higher running rate due to the small number of Transformer blocks that are used in a serial CNN-Transformer. DenseFuse, RFN-Nest, U2Fusion, SeAFusion, and UNFusion utilize convolution for local feature encoding and decoding with less computational complexity. Whereas ConvNeXt blocks are employed as the main encoding backbone in DTFusion, it is relatively time-consuming because the inference throughputs of ConvNeXts are comparable to or exceed that of Swin Transformers, but it is much better than CSF, GANMcC, and SMoA. Furthermore, by combining the comparative results of multiple fusion methods, our method achieves competitive operational efficiency.

## 5. Conclusions

In this article, a novel end-to-end framework based on PConv-ConvNeXt and texture-contrast compensation is proposed for infrared and visible image fusion. In the framework, we construct a dense encoder network with large kernel-size convolutions to efficiently extract features with larger receptive fields. In addition, gradient residuals and attention mechanisms are also applied to compensate for the texture details and contrast of features. The excellent visual effect and higher evaluation metrics are implemented on the MSRS, TNO, and M${}^{3}$FD datasets to show that the proposed method outperforms the other nine state-of-the-art fusion methods.

## Figures and Tables

**Figure 1 sensors-24-00203-f001:**
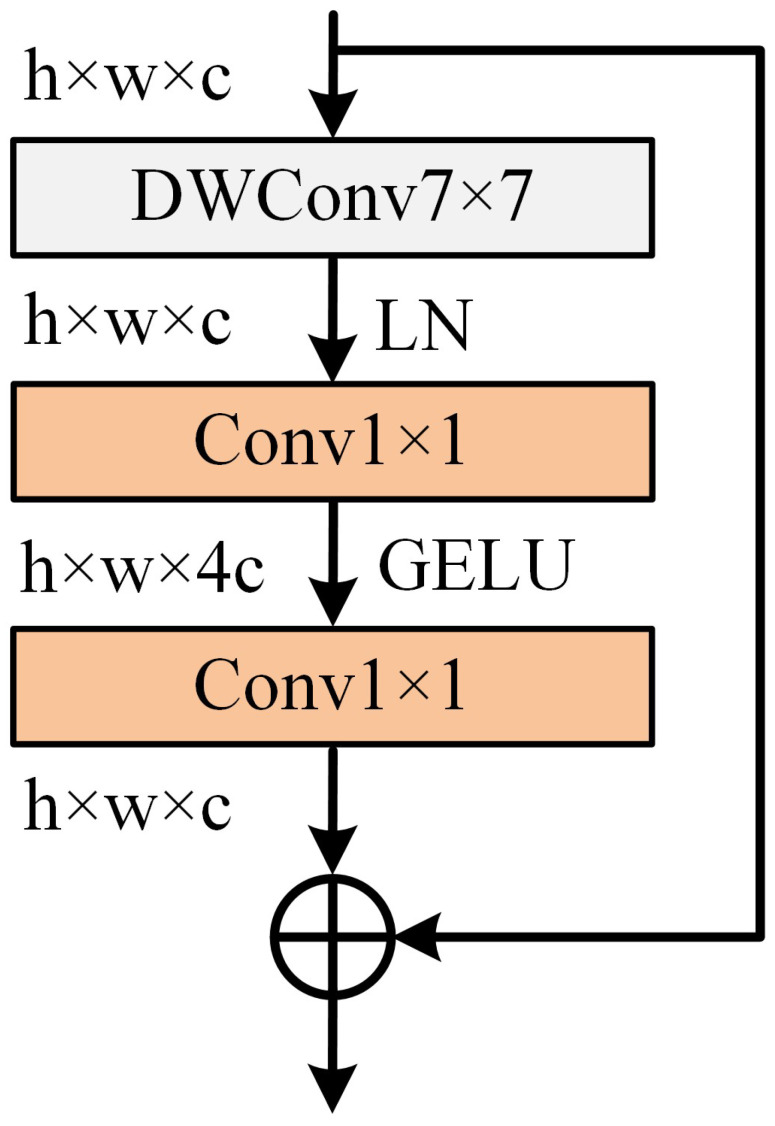
The structure of the ConvNeXt block.

**Figure 2 sensors-24-00203-f002:**
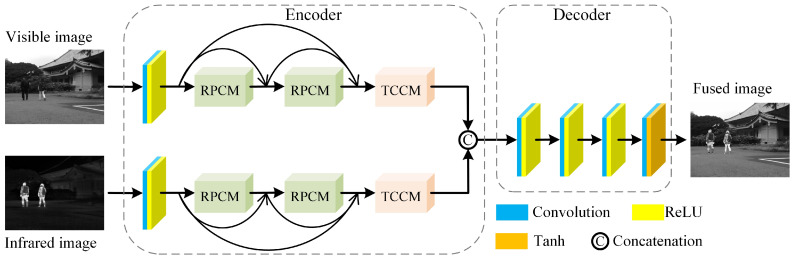
The overall processes of our framework. Two independent branches are designed to extract the features from different modalities. In each branch, RPCM is used to extract features with larger receptive fields, and dense connections are applied between RPCMs to reuse intermediate features. Then, TCCM is used to compensate for texture details and contrast of features.

**Figure 3 sensors-24-00203-f003:**
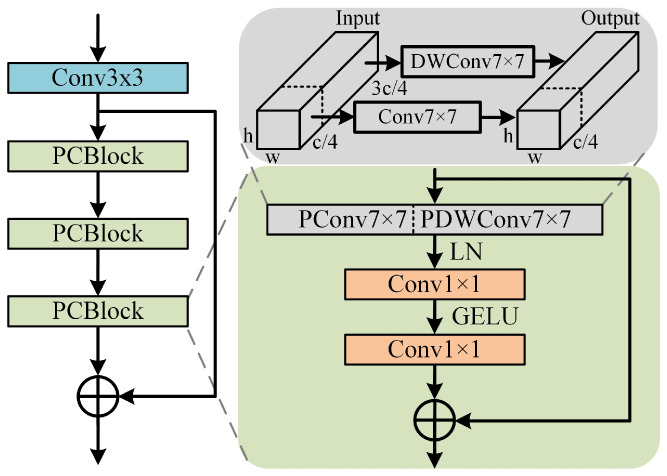
The structure of RPCM. It consists of a convolutional layer and residual connections of three PCBs. Each PCB is constructed by integrating a 7 × 7 regular convolution into the first layer of the ConvNeXt block.

**Figure 4 sensors-24-00203-f004:**
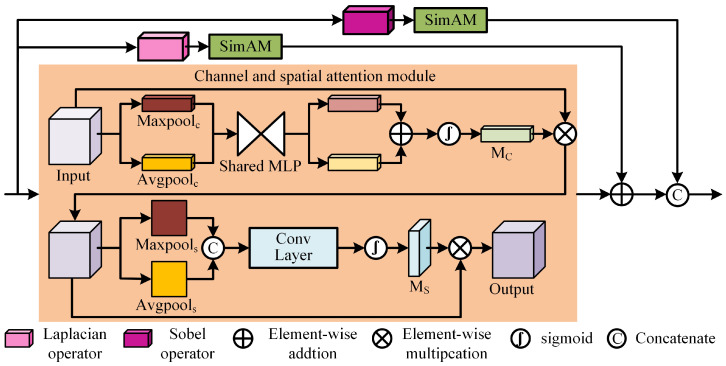
The architecture of TCCM. TCCM consists of two gradient residuals with SimAM attention and a CSAM, where the specific implementation of the CSAM is indicated in the orange rectangle.

**Figure 5 sensors-24-00203-f005:**
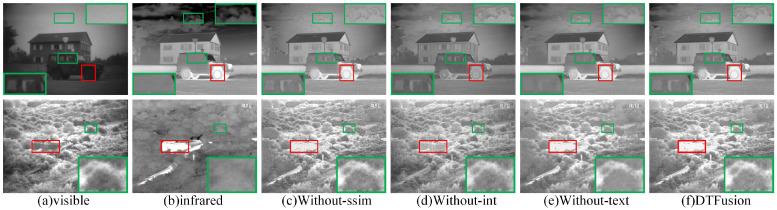
Ablation analysis of loss function on the TNO dataset.

**Figure 6 sensors-24-00203-f006:**
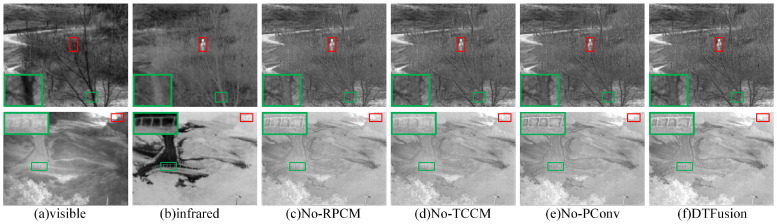
Ablation analysis of the network structure on the TNO dataset.

**Figure 7 sensors-24-00203-f007:**
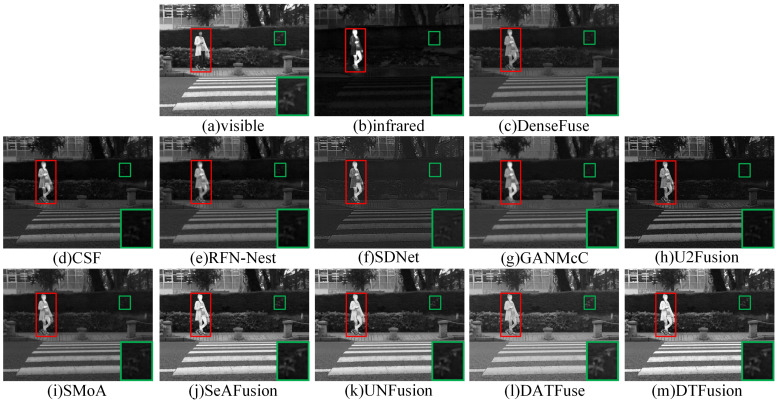
Subjective comparison of different fusion methods on 00123D from the MSRS dataset.

**Figure 8 sensors-24-00203-f008:**
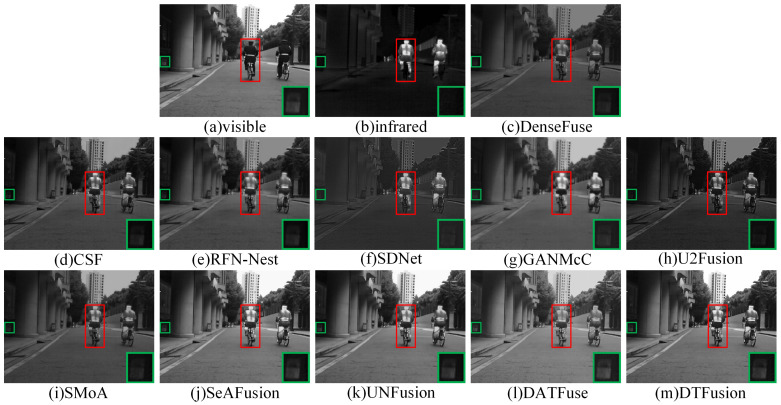
Subjective comparison of different fusion methods on 00537D from the MSRS dataset.

**Figure 9 sensors-24-00203-f009:**
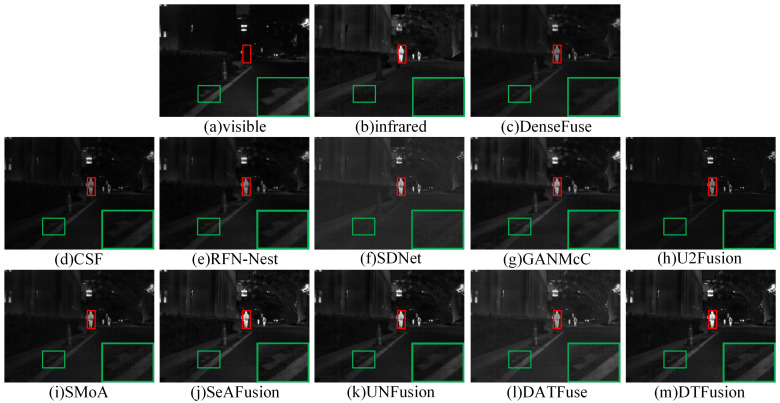
Subjective comparison of different fusion methods on 00858N from the MSRS dataset.

**Figure 10 sensors-24-00203-f010:**
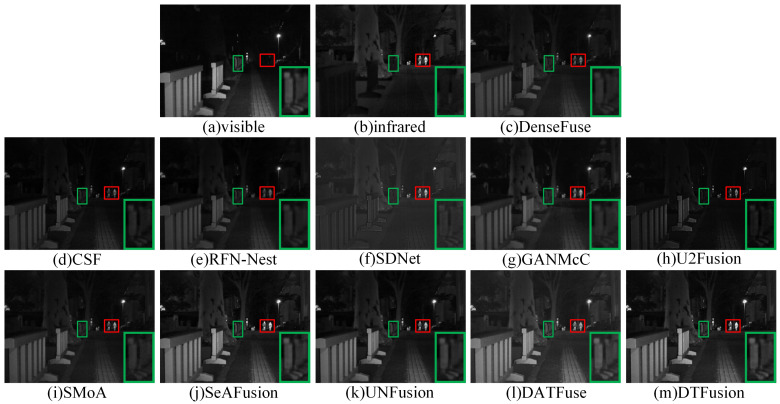
Subjective comparison of different fusion methods on 01024N from the MSRS dataset.

**Figure 11 sensors-24-00203-f011:**
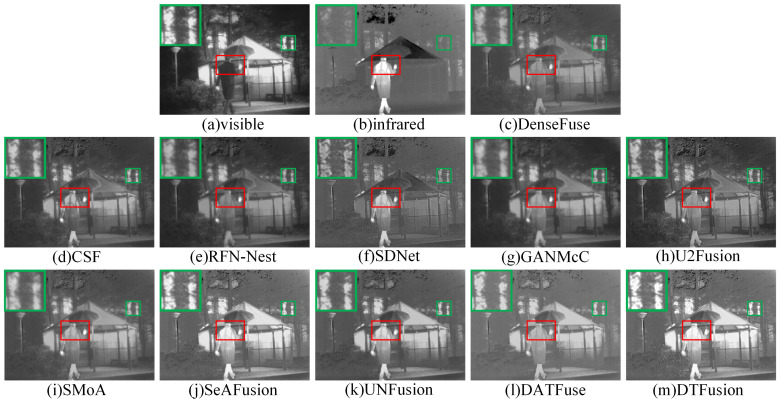
Subjective comparison of different fusion methods on *Kaptain_1654* from the TNO dataset.

**Figure 12 sensors-24-00203-f012:**
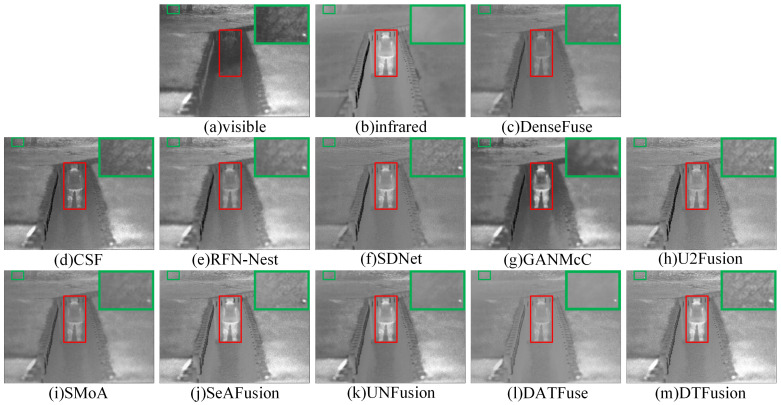
Subjective comparison of different fusion methods on *soldier_in_trench_2* from the TNO dataset.

**Figure 13 sensors-24-00203-f013:**
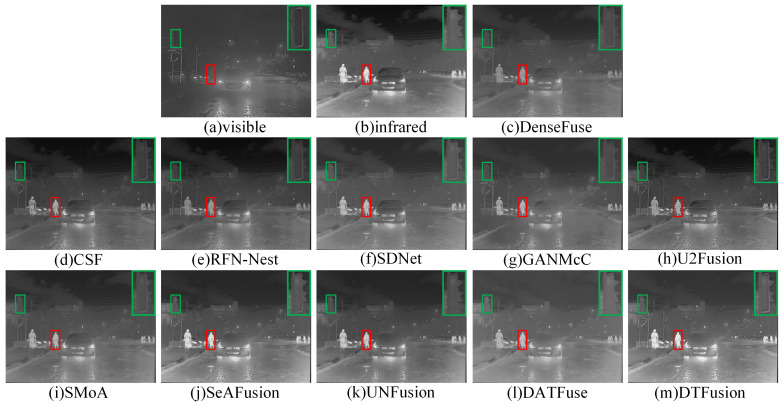
Subjective comparison of different fusion methods on a representative example from the M${}^{3}$FD dataset.

**Table 1 sensors-24-00203-t001:** Ablation study of loss function on the TNO dataset (the best results are in bold, and the second-best results are underlined).

Loss Function	EN	SF	AG	SD	SCD	VIF	Q${}_{\mathrm{abf}}$	SSIM	MS-SSIM
Without-ssim	7.0570	**12.7583**	**4.9627**	44.8685	1.6374	**0.7672**	**0.5999**	1.0168	1.0257
Without-int	6.9578	11.0307	4.3695	39.4983	1.7283	0.6511	0.5380	0.9954	1.0785
Without-text	6.9374	8.9991	3.4665	43.0512	1.6836	0.6661	0.4540	1.0434	1.0762
DTFusion	**7.0729**	12.6585	4.9350	**45.8861**	**1.7614**	0.7199	0.5660	**1.0445**	**1.0788**

**Table 2 sensors-24-00203-t002:** Ablation study of the network structure on the TNO dataset (the best results are in bold, and the second-best results are underlined).

Models	EN	SF	AG	SD	SCD	VIF	Q${}_{\mathrm{abf}}$	SSIM	MS-SSIM
No-RPCM	7.0417	12.6129	4.8514	44.4450	1.6988	**0.7598**	0.5863	1.0539	1.0628
No-PConv	7.0563	12.6217	**4.9650**	44.6881	1.7133	0.7457	**0.5947**	1.0371	1.0683
No-TCCM	7.0063	12.0825	4.6448	43.1418	1.6482	0.7363	0.5645	**1.0697**	1.0648
DTFusion	**7.0729**	**12.6585**	4.9350	**45.8861**	**1.7614**	0.7199	0.5660	1.0445	**1.0788**

**Table 3 sensors-24-00203-t003:** Objective comparison of different fusion methods on the MSRS dataset (the best results are in bold, and the second-best results are underlined).

Models	EN	SF	AG	SD	SCD	VIF	Q${}_{\mathrm{abf}}$	SSIM	MS-SSIM
DenseFuse	5.9455	6.0676	2.0973	23.5937	1.2519	0.6845	0.3651	0.9019	1.0070
CSF	5.8701	7.1226	2.3864	26.6909	1.3490	0.6119	0.3697	0.7672	0.9813
RFN-Nest	6.2102	6.1924	2.1460	29.1218	1.4717	0.6495	0.3873	0.7604	1.0090
SDNet	5.2526	8.6887	2.7218	17.3375	0.9880	0.4948	0.3726	0.7322	0.8932
GANMcC	6.2296	5.9199	2.2062	28.2696	1.4816	0.6273	0.3298	0.8082	0.9734
U2Fusion	5.3743	9.0844	2.8687	25.5003	1.2428	0.5297	0.4076	0.7635	0.9099
SMoA	6.1164	7.5615	2.5667	28.8636	1.3884	0.6566	0.4339	0.8599	1.0094
SeAFusion	**6.7031**	11.5627	3.8788	43.0600	1.7014	**0.9749**	**0.6653**	0.9759	1.0363
UNFusion	6.5652	10.4136	3.3311	40.9665	1.6590	0.9456	0.6513	0.9984	1.0476
DATFuse	6.5043	10.9776	3.6513	36.5945	1.4092	0.9023	0.6394	0.9060	1.0046
DTFusion	6.6331	**12.6396**	**4.0827**	**44.1737**	**1.7358**	0.9200	0.6550	**1.0199**	**1.0685**

**Table 4 sensors-24-00203-t004:** Objective comparison of different fusion methods on *Kaptain_1654* from the TNO dataset (the best results are in bold, and the second-best results are underlined).

Models	EN	SF	AG	SD	SCD	VIF	Q${}_{\mathrm{abf}}$	SSIM	MS-SSIM
DenseFuse	6.3276	6.3882	2.5563	27.3346	1.5923	0.5736	0.3268	1.0078	0.8233
CSF	6.5561	7.2109	3.0127	32.7923	1.7185	0.5393	0.3621	0.9734	0.8851
RFN-Nest	6.5620	5.0514	2.1920	31.7220	1.6656	0.5214	0.2983	0.7364	0.8402
SDNet	6.2167	10.9474	4.3723	24.4714	1.5514	0.4706	0.3880	1.0114	0.8498
GANMcC	6.6511	5.4166	2.3573	35.3765	1.7362	0.5031	0.2565	0.8921	0.8195
U2Fusion	**6.8309**	10.2560	4.4312	34.0316	1.7381	0.5588	0.4257	1.0221	0.9091
SMoA	6.4447	6.3792	2.6192	32.7428	1.6986	0.5007	0.3122	0.7795	0.8306
SeAFusion	6.7105	11.2986	4.6320	41.4455	1.7177	0.5796	0.4512	0.8727	0.8911
UNFusion	6.4740	8.7765	3.3405	36.8532	1.6765	**0.7865**	0.5305	**1.0426**	0.8576
DATFuse	6.2341	9.5534	3.6206	29.7492	1.4967	0.7020	0.5107	0.9582	0.7725
DTFusion	6.7212	**12.4824**	**4.8506**	**42.7354**	**1.7761**	0.6860	**0.5579**	0.9889	**0.9248**

**Table 5 sensors-24-00203-t005:** Objective comparison of different fusion methods on *soldier_in_trench_2* from the TNO dataset (the best results are in bold, and the second-best results are underlined).

Models	EN	SF	AG	SD	SCD	VIF	Q${}_{\mathrm{abf}}$	SSIM	MS-SSIM
DenseFuse	6.3070	7.5459	2.7402	20.0567	1.5948	0.6927	0.3953	0.8886	0.9066
CSF	6.9345	8.8690	3.8940	31.7183	1.7707	0.6516	0.5417	0.9249	0.9967
RFN-Nest	7.0609	7.4689	3.2704	33.2321	**1.8660**	0.6336	0.5001	0.8497	**0.9975**
SDNet	6.0010	**14.7995**	5.4140	19.1660	1.5612	0.6632	0.6035	**0.9788**	0.8821
GANMcC	**7.0623**	7.0105	2.7162	**34.8298**	1.8269	0.6139	0.3044	0.8123	0.9479
U2Fusion	6.5952	13.0062	5.2435	24.9478	1.7591	0.7047	0.5988	0.9762	0.9933
SMoA	6.4538	7.0206	2.3414	22.7674	1.6671	0.5060	0.2235	0.6436	0.9036
SeAFusion	6.7607	13.2970	4.9708	30.0178	1.7232	0.5967	0.4465	0.8141	0.9498
UNFusion	6.6805	11.3169	3.7477	29.6441	1.6904	**0.9285**	0.5403	0.9761	0.9639
DATFuse	6.0989	9.2577	2.9807	20.1861	1.4767	0.6958	0.4169	0.8643	0.8581
DTFusion	6.7794	13.7397	**5.4402**	31.0218	1.7768	0.7701	**0.6868**	0.9435	0.9543

**Table 6 sensors-24-00203-t006:** Objective comparison of different fusion methods on 20 image pairs from the TNO dataset (the best results are in bold, and the second-best results are underlined).

Models	EN	SF	AG	SD	SCD	VIF	Q${}_{\mathrm{abf}}$	SSIM	MS-SSIM
DenseFuse	6.5312	7.1421	2.7540	28.2952	1.5355	0.5997	0.3581	1.0187	1.0348
CSF	6.9622	9.0222	3.7094	39.1585	1.7211	0.5920	0.4140	0.9579	**1.0824**
RFN-Nest	6.9988	6.1282	2.7297	38.4950	1.7179	0.5640	0.3373	0.8064	1.0476
SDNet	6.6880	12.0056	4.6001	34.1692	1.4700	0.5747	0.4447	1.0092	1.0253
GANMcC	6.9580	6.7637	2.7919	38.8544	1.6641	0.5450	0.3079	0.8900	1.0260
U2Fusion	6.9716	11.8527	4.8584	38.6913	1.7087	0.6076	0.4494	0.9835	1.0790
SMoA	6.7095	7.3701	2.8099	33.3535	1.6411	0.5230	0.3291	0.8108	1.0424
SeAFusion	**7.1809**	12.4793	4.9169	**47.1625**	1.7248	0.6138	0.4488	0.8744	1.0467
UNFusion	7.0402	10.5738	3.9037	44.0714	1.6746	**0.8613**	0.5580	1.0430	1.0609
DATFuse	6.6465	10.7138	3.9266	31.7927	1.4900	0.7049	0.5083	0.9507	0.9679
DTFusion	7.0729	**12.6585**	**4.9350**	45.8861	**1.7614**	0.7199	**0.5660**	**1.0445**	1.0788

**Table 7 sensors-24-00203-t007:** Objective comparison of different fusion methods on 300 image pairs from the M${}^{3}$FD dataset (the best results are in bold, and the second-best results are underlined).

Models	EN	SF	AG	SD	SCD	VIF	Q${}_{\mathrm{abf}}$	SSIM	MS-SSIM
DenseFuse	6.4268	8.057	2.8312	24.9986	1.5358	0.5955	0.3793	0.9416	1.0463
CSF	6.8299	10.2153	3.7065	33.8474	1.7298	0.6197	0.4788	0.9340	1.0825
RFN-Nest	6.8271	8.2239	3.0637	32.8573	1.7279	0.5745	0.4155	0.8007	1.0444
SDNet	6.8481	14.3417	5.0301	35.6101	1.5508	0.5613	0.5372	0.9690	0.9391
GANMcC	6.8482	7.7419	2.7907	33.3603	1.6540	0.5359	0.3195	0.8452	0.9873
U2Fusion	6.9532	13.8198	5.1340	35.2004	**1.7635**	0.6612	0.5668	0.9696	**1.0835**
SMoA	6.6054	8.0088	2.7677	28.6175	1.6261	0.4939	0.3015	0.6419	0.9655
SeAFusion	**7.0037**	15.3494	5.4669	37.1395	1.6843	0.5821	0.5152	0.7518	0.9897
UNFusion	6.8598	12.1175	4.0451	34.4041	1.5702	0.7594	0.5477	**1.0049**	1.0344
DATFuse	6.4019	10.4622	3.4408	26.3103	1.2867	0.6444	0.4935	0.9193	0.9343
DTFusion	6.9707	**17.1730**	**5.8161**	**37.5127**	1.6856	**0.7615**	**0.6548**	0.9506	1.0432

**Table 8 sensors-24-00203-t008:** Comparison of running time of different methods on the MSRS, TNO, and M${}^{3}$FD datasets (the best results are in bold, and the second-best results are underlined).

Models	MSRS (s)	TNO (s)	M${}^{3}$FD (s)
DenseFuse	0.0778	0.0864	0.1896
CSF	32.2675	31.9109	77.0992
RFN-Nest	0.1361	0.2166	0.2613
SDNet	0.0239	**0.0153**	0.0569
GANMcC	5.4610	5.4283	13.2267
U2Fusion	0.1897	0.1828	0.5035
SMoA	8.2952	8.6730	9.1976
SeAFusion	0.1307	0.5514	0.2711
UNFusion	0.1102	0.2673	0.2491
DATFuse	**0.0203**	0.0269	**0.0429**
DTFusion	0.3765	0.5447	0.8927

## Data Availability

The authors confirm that the data supporting the findings of this study are available within the article.

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
