# Peer review of "DTFusion: Infrared and Visible Image Fusion Based on Dense Residual PConv-ConvNeXt and Texture-Contrast Compensation"

_sensors, 2023, doi:10.3390/s24010203_

Round 1

Reviewer 1 Report

Comments and Suggestions for Authors

The paper proposes an end-to-end framework for infrared and visible image fusion.

The framework has the following features:

1. used ConvNeXt blocks for feature extraction, and used the partial convolution to make feature extraction efficient.

2. dense connections were applied between convolution modules to reuse intermediate features.

3. a texture-contrast compensation module (TCCM) with gradient residuals and attention mechanism was used to compensate for texture details and contrast of features.

The paper is well-written and easy to understand what the proposed method is and how it works.

However, I cannot recommend that the paper be accepted for the following reasons.

There is no review of the latest studies (published in 2022, 2023) related to visible-IR image fusion.

In related work, a review of transformer-based methods or fusion (CNN + Transformer) methods for visible-IR image fusion should be added.

In experimental results, performance comparisons with the latest studies (including CNN-based, Transformer-based) should be added.

The methods and models proposed in this study are those already introduced and utilized in existing studies related to deep learning-based vision tasks. In short, I'm not sure what's new in the proposed framework.

There is no clear or impressive performance improvement in the experimental results compared to the existing methods. The latest methods have not been compared.

On line 202, the authors mentioned, "For the fused image, the structural information from the two source images is equally important. Therefore, we set the parameter scale w1 = w2 = 0.5." However, two source images can be of different importance depending on image quality or use, requiring an adaptive method of determining the weights. 

In Fig. 5 and Table 1, the results of "Without-ssim" look similar to or better than the proposed method. It indicates that the proposed method is sensitive to the hyper-parameters (lambdas). Please elaborate this issue.

In Fig. 4, please explain why the Sobel gradients are concatenated while the Laplacian gradients are summed.

Please share the source code so that the proposed model is available for public use.

Reviewer 2 Report

Comments and Suggestions for Authors

A novel end-to-end infrared and visible image fusion framework called DTFusion is proposed to address these problems. A residual PConv-ConvNeXt module (RPCM) and dense connections are introduced into the encoder network to efficiently extract features with larger receptive fields. In addition, a texture-contrast compensation module (TCCM) with gradient residuals and attention mechanism is designed to compensate for the texture details and contrast of features. The fused features are reconstructed through four convolutional layers to generate a fused image with rich scene information. Experiments on public datasets show that DTFusion outperforms other state-of-the-art fusion methods in both subjective vision and objective metrics. However, the following problem need to be improved.

(1) The English of paper is poor, please polish it again.

(2) Each variable in the equation should be explained clearly.

(3) Why use PConv?

(4) Please provide all parameters of the proposed model.

(5) More recently references should be cited, such as

[*]Image fusion based on complex-shearlet domain with guided filtering. Multidimensional Systems and Signal Processing, 2017.  

[**]Infrared and visible image fusion based on Multi-State contextual hidden Markov Model.Pattern Recognition, 2023.

Comments on the Quality of English Language

NA

Reviewer 3 Report

Comments and Suggestions for Authors

Positive:
- appropriate analysis in the introduction,
- extensive experiments and comparison with various techniques,
- appropriately selected metrics for validation,
- an innovative solution for image fusion based on ConvNeXt block.

To improve:
- There is no comparison of speed results. It is not known whether the proposed method is faster or slower than the others (the qualitative results are very similar to the referenced works and there is no clear advantage).
- Captions under the charts are sometimes illegible: it would be useful to add a description for each method under the image.
- Charts 5 and 6 with their captions are not entirely understandable

Some detailed comments:

Line 233: "MSRS dataset [41,42] containing 1083 training image pairs and 361 test image pairs was used for training and evaluation of our fusion task."
It is not entirely clear how many images there were for training, how many for validation and how many for testing for individual experiments (and whether the sets were disjoint or not).

Comments on the Quality of English Language

Line 244: "And the Pytorch platform is used for the proposed program." Do not start the sentense from 'And'.

Round 2

Reviewer 1 Report

Comments and Suggestions for Authors

The paper has been properly revised according to the comments. I think that it can be accepted in its current form.

Reviewer 2 Report

Comments and Suggestions for Authors

The respond is OK.